# Antioxidant, Antidiabetic, and Vasorelaxant Effects of Ethanolic Extract from the Seeds of *Swietenia humilis*

**DOI:** 10.3390/ijms26052063

**Published:** 2025-02-27

**Authors:** Elizabeth Alejandrina Guzmán Hernández, Gladys Chirino Galindo, Rubén San Miguel Chávez, Patricia Castro Moreno, Maximiliano Ibarra Barajas, Tomás Ernesto Villamar Duque, Anayantzin Paulina Heredia Antúnez, Leonardo del Valle Mondragón, Gil Alfonso Magos Guerrero, David Segura Cobos

**Affiliations:** 1Medical Surgeon Career, Faculty of Higher Studies Iztacala, National Autonomous University of Mexico, Tlalnepantla C.P. 54090, Mexico; seguracd@unam.mx; 2Diabetes Mellitus Metabolism Laboratory, Faculty of Higher Studies Iztacala, National Autonomous University of Mexico, Tlalnepantla C.P. 54090, Mexico; gchrinino@hotmail.com; 3Phytochemistry Area, Postgraduate in Botany, Postgraduate College, Mexico City 11340, Mexico; sanmi@colpos.mx; 4Biomedicine Unit, Faculty of Higher Studies Iztacala, National Autonomous University of Mexico, Tlalnepantla C.P. 54090, Mexico; patriciac@iztacala.unam.mx (P.C.M.); maxibarrab@hotmail.com (M.I.B.); 5General Bioterium, Faculty of Higher Studies Iztacala, Biology, National Autonomous University of Mexico (UNAM), Tlalnepantla C.P. 54090, Mexico; vidutoer@yahoo.com.mx (T.E.V.D.); paulina_852@hotmail.com (A.P.H.A.); 6Department of Pharmacology, National Institute of Cardiology Ignacio Chavez, Mexico City 04510, Mexico; leonardodvm65@hotmail.com; 7Department of Pharmacology, Faculty of Medicine, University National Autonomous of Mexico (UNAM), Mexico City 04510, Mexico; gamagos@unam.mx

**Keywords:** *Swietenia humilis*, antihypertensive plants, antidiabetic plant

## Abstract

Arterial hypertension and diabetes mellitus are components of the cardiometabolic syndrome that arises from a sedentary lifestyle, excess calorie intake, and obesity. Swietenia humilis Zucc has been used in traditional Mexican medicine for the treatment of diabetes mellitus; this work investigated the antioxidant, antidiabetic, and vasorelaxant effects of ethanolic extract of S. humilis seeds. The phytochemical composition of the extract was analyzed by high-performance liquid chromatography. To study the hypoglycemic effect, the activity of antioxidant enzymes (catalase, superoxide dismutase, and glutathione peroxidase) and markers of oxidative stress (malondialdehyde and 8-hydroxy-2-desoxyguanosine) were evaluated in the model of diabetes mellitus induced by nicotinamide and streptozotocin in rats. The vasodilatory effect of the extract was tested in rat aortic rings. The ethanolic extract of seeds of Swietenia humilis showed antioxidant, hypoglycemic, and endothelium-independent vasorelaxant effects, probably by blocking calcium transport, likely due to ursolic acid and α-amyrin, phytochemical compounds more abundant in the extract.

## 1. Introduction

According to the International Diabetes Federation, Mexico ranks seventh worldwide in cases of diabetes mellitus (IDF, 2021) [1]; data from the National Institute of Public Health indicate that the prevalence is 18.3%, with 22.1% associated with prediabetes [2]. Data from INEGI, 2024 mention that it remains the second cause of death in the Mexican population [3,4]. There are multiple mechanisms involved in the development of diabetes mellitus; however, it has been shown that oxidative stress plays an important role in the evolution of the disease and its association with macro- and microvascular complications.

Among the microvascular complications, more than 40% of diabetic patients have chronic renal failure due to diabetic nephropathy, which is one of the main causes of mortality in type 1 diabetic patients. Hyperglycemia is considered one of the main triggers of this process because it can exert harmful effects on mesangial, tubular, interstitial, and vascular cells through the accumulation of advanced glycation products, the activation of the polyol pathway via hexosamine, the activation of the renin angiotensin system, and the generation of oxidative stress; under pathological conditions, the balance of the production and detoxification of reactive oxygen species is lost due to a failure in the activity of antioxidant enzymes.

The excessive production of reactive oxygen species causes damage to biomolecules such as lipids, proteins, and DNA; in endothelial cells, it can increase vascular permeability through modifications in the translation of signals that nitric oxide synthesis mainly decreases, which causes the activity of the renin–angiotensin system to increase, causing endothelial dysfunction [5].

Angiotensin II activates nicotinamide adenine dinucleotide phosphate (NADPH) oxidase (NOX), which produces a superoxide anion, thereby promoting renal vascular remodeling and increasing preglomerular resistance [6]. Protein kinase C promotes significant upregulation of NOX4 and enhances the generation of oxygen species [7]; in particular, the superoxide anion that is generated reacts with nitric oxide to form peroxynitrite, which participates in cellular protein damage, lipid peroxidation, and DNA damage, by degrading nitric oxide synthesis more quickly. In endothelial cells, important changes are also established since stopping the production of nitric oxide can lead to endothelial dysfunction, resulting in arterial hypertension and the progression of kidney damage, as well as angiotensin II [7]. To counteract these harmful effects, the cell has antioxidant defense mechanisms, among which are antioxidant enzymes, such as superoxide dismutase (SOD), catalase (CAT), and glutathione peroxidase (GPX); in diabetic nephropathy, the activity of these enzymes is decreased [8,9].

In vitro and in vivo studies have shown that the intake of natural antioxidants is realized, mainly, through a diet that includes fruits, vegetables, and medicinal plants. They contain secondary metabolites such as phenolic compounds, among which phenolic acids, tannins, lignins, and flavonoids have diverse antioxidant mechanisms of action and can be mediated by oxidation-reduction reactions or by free radical scavenging; for example, resveratrol, curcumin, quercetin, baicalin, naringenin, silibinin, apigenin, luteolin, chlorogenic acid, elagic acid, and caffeic acid, administered in an adequate dosage regimen, can be beneficial for the restoration of antioxidant systems [10].

*Swietenia humilis* Zucc. is known in the State of Sinaloa, Mexico, as venadillo, zopilote, caobilla, or cubano. It is in the dry and humid tropical forests of Pacific coasts in the States of Sinaloa, Colima, Michoacán, Guerrero, and Chiapas in Mexico [11] for treating different diseases, such as stomach pain, and amoebic dysentery. The decoction of the seed is consumed in the treatment of the inflammation of the intestines and kidneys [12]. After having been isolated and identified from *S. humilis* tetranortriterpenoids of the mexicanolide group, humulin B, methyl-2-hydroxy-3-b-isobutyroxy-1-oxomeliac-8(30)-enate, methyl-2-hydroxy-3-b-tigloyloxy-1-oxomeliac-8(30)-enate, humilinolide C, 2-hydroxydestigloyl-6-deoxyswietenine acetate, methyl-2-hydroxy-3-β-tigloyloxy-1-oxomeliac-8(30)-enate, and humilinolide H, the 2-hydroxy-destigloyl-6-deoxyswietenine acetate (mexicanolide 1) has been reported to have antidiabetic activity [12]. This work aimed to study the antioxidant, nephroprotective, and hypotensive effects of ethanolic extract of seeds from *Swietenia humilis*.

## 2. Results

### 2.1. Phytochemical Profile of Ethanolic Extract of S. humilis

Figure 1 and Table 1 show the results obtained from the phytochemical analysis of the ethanolic extract of *S. humilis*, with the most abundant being α-amyrin (68%), galangin (1.25%), myricetin (1.25%), floretin (1.67%), and ursolic acid (12.4%).

### 2.2. Antioxidant Activity In Vitro S. humilis

For the total phenol content of the ethanolic extract of S. humilis of 76 mg/g of gallic acid equivalents, the total flavonoid concentration of 18.15 mg/g of quercetin equivalents was obtained, along with the average antioxidant concentration (CA50), which is the concentration of extract or quercetin that reduces the 2,2-diphenyl-1-picrylhydrazyl radical by 50%, and the ethanolic extract of S. humilis was 35%.

### 2.3. Whole Animal Data

Table 2 shows whole animal data for control (C), diabetes mellitus (DM), diabetes mellitus + glibenclamide (DM + Gli), and diabetes mellitus + ethanolic extract of *S. humilis* (DM + EtOH): glycemia, body weight, food and water consumption, and urinary volume. In untreated diabetes mellitus, plasma glucose was 480 ± 19 mg/dL compared with the normoglycemic control group (100 ± 8 mg/dL, *p* < 0.05). DM + glibenclamide and DM + EtOH 100 and 200 mg/kg decreased hyperglycemia without modifying water consumption and urinary volume (Table 2).

### 2.4. Effect of Ethanolic Extract of S. humilis on Malondialdehyde, Malonate, and 8-Hydroxy-2-Deoxyguanosine

As can be seen in Figure 1, the total antioxidant activity decreased in the diabetic rat group, as markers of oxidative stress from lipoperoxidation were quantified through malondialdehyde and DNA damage (8-hydroxy-2-desoyguanosine) and were significantly increased in the diabetes mellitus group compared to the normoglycemic control.

The ethanolic extract of *S. humilis* with the dose of 100 and 200 mg/kg treatment produces a significant reduction in malondialdehyde and 8-hydroxy-2-desoxyguanosine in the diabetes mellitus group, restoring total antioxidant activity.

### 2.5. Effect of Ethanolic Extract of S. humilis on Antioxidant Activity in the Renal Cortex

In the renal cortex, the antioxidant activities of catalase, superoxide dismutase, and glutathione peroxidase significantly decreased in the diabetic rats group compared to the normoglycemic control. In the groups treated with ethanolic extract of *S. humilis,* at doses of 100 or 200 mg/kg, the activity of the enzyme catalase and superoxide dismutase increased, only increasing with the dose of 200 mg/kg of glutathione peroxidase (Figure 2).

### 2.6. Nephroprotective Effect of S. humillis

In Figure 3a, the normal morphological architecture was observed as follows: the kidney cortex was located externally, towards the periphery, and had a labyrinthine shape, and the proximal and distal convoluted tubules had a normal appearance. In Figure 3b, the induction of diabetes mellitus causes great damage, the histology of the renal cortex is abnormal, and there are crescents, as well as the loss of glomeruli, proliferation of cells and glomerular mesangial matrix, and dilation of the Bowman space. Regarding the treatment with glibenclamide (Figure 3c) and the ethanolic extract (Figure 3d) of *S. humillis* at a dose of 200 mg/kg, the appearance of the glomeruli was like the normoglycemic control group alone.

### 2.7. Effect of Ethanolic Extract of S. humilis on Plasma Angiotensin II, Angiotensin 1-7, and Nitric Oxide

As indicators of vascular damage, the vasoactive peptides plasma angiotensin II, plasma angiotensin 1-7, and nitric oxide in diabetes mellitus induced with streptozotocin/nicotinamide were quantified. In Figure 4, it can be observed that angiotensin II and nitric oxide increased in the group of diabetic rats compared to the control normoglycemic group. Angiotensin 1-7 is not modified; however, the presence of the ethanolic extract of *S. humilis* decreased the synthesis of nitric oxide with doses of 100 and 200 mg/kg (Figure 4).

### 2.8. Effects of Ethanolic Extract of S. humilis on High K^+^- and NA-Induced Vascular Contractions

In rat isolated aortic rings, the maximal tension obtained with 80 mM high K^+^ and 10^−5^ M phenylephrine indicated 100% contraction (*n* = 3) relative to basal tension. As shown in Figure 5a, the cumulative dosing of the ethanolic extract of *S. humilis* (EtOH) produced concentration-dependent relaxation of the phenylephrine-precontracted aortic rings without endothelium EtOH, relaxing the phenylephrine-induced vascular contraction; with a half-maximal effective concentration (EC_50_) of 0.04 mg/mL, and relaxing the high K^+^-induced vascular contraction, with an EC_50_ of 0.0153 mg/mL (Figure 5b). In the endothelium-intact aortic rings, EtOH relaxed the phenylephrine-induced vascular contraction with an EC_50_ of 0.11 mg/mL (Figure 5a) and relaxed the high K^+^-induced vascular contraction with an EC_50_ of 0.014 mg/mL.

### 2.9. Ethanolic Extract of S. humilis-Promoted Vasorelaxation by L-Type Calcium Channels

The experimental results of vascular aortic rings pre-incubated with L-NAME showed that the ethanolic extract of *S. humilis* could still relax blood vessels in a concentration-dependent manner after endothelial nitric oxide synthase was inhibited. In the endothelium-intact aortic rings, EtOH relaxed with an EC_50_ of 0.136 mg/mL; without endothelium, the EC_50_ obtained was 0.078 mg/mL. Therefore, the ethanolic extract of *S. humilis* may relax blood vessels mainly by directly acting on vascular smooth muscle (Figure 6).

To determine the role of voltage-gated calcium channels in the ethanolic extract of *S. humilis*-induced vasorelaxation, the aortic rings were pre-incubated with Krebs 80 mM without Ca^2+^ (Figure 7).

## 3. Discussion

The results of the present study showed that the ethanolic extract of seeds of *Swietenia humilis* had antioxidant, hypoglycemic, and endothelium-independent vasorelaxant effects, probably by blocking calcium transport, likely due to ursolic acid and α-amyrin, of which phytochemical compounds are more abundant in the extract.

In this study, a type 2 diabetes model was used through the administration of nicotinamide and streptozotocin. Its cytotoxic action of streptozotocin is mainly measured by the excessive production of reactive oxygen species that causes the apoptosis of pancreatic beta cells and the attenuation of insulin synthesis. When the administration is followed by nicotinamide, it partially protects the pancreatic beta cell from destruction, thus generating a type 2 diabetes mellitus model in which moderate hyperglycemia, impaired glucose tolerance, insulin resistance, polydipsia, polyphagia, and polyuria can be observed [13].

In the present study, the phytochemical components in the ethanolic extract of *S. humilis* contain α-amyrin and ursolic acid as major components, and previous studies have shown that they have antihyperglycemic activity; however, other secondary metabolites were also detected in a lower percentage such as quercetin and myricetin. Previous studies suggest that they improve insulin resistance and activate adenosine monophosphate kinase, a mechanism of action similar to that of metformin [14,15]. Previous studies on ferulic acid suggest a secretagogue action [16], and α-amyrin has been shown to inhibit alpha-glucosidase [17].

For the diagnosis of diabetes mellitus, the presence of polyphagia, polydipsia, polyuria, and weight loss are taken into account when observed in the present study. It is worth mentioning that, in the presence of the ethanolic extract of *S. humilis*, weight gain was observed; concerning the group of diabetic rats without treatment, it could be due to the presence of quercetin and myricetin, which improve the secretion and action of insulin [14,15]. During the initial phase of diabetes mellitus induced by the administration of streptozotocin, the generation of oxidative stress increases, which may be lower in the presence of nicotinamide since it protects the pancreatic beta cell from its total destruction. Moreover, the generated reactive oxygen species can interact with lipids, proteins, and DNA, causing cellular damage. There are markers of oxidative stress such as malonedialdehyde, which is a byproduct of the peroxidation of polyunsaturated fatty acids in cells, and 8-hydroxy-2-deoxyguanosine, a marker of DNA damage that increased in activity in diabetes mellitus [18]. To counteract this oxidative damage, the body has enzymatic systems that help counteract this harmful effect, of which catalase, superoxide dismutase, and glutathione peroxidase are the most important. In diabetes mellitus, this enzymatic system decreased, which is in accordance with the results obtained in the present study. The ethanolic extract of *S. humilis* possessed high phenolic contents (76 mg GAE/g of extract) and can directly neutralize the free radicals generated by lipid peroxidation compared to other extracts from other plants, including *Buddleja cordata*, of which the ethanol extract showed 157 mgEq gallic acid/g [19], and *Chiranthodendron pentadacylon Larreat* (221.4 mg GAE/g of extract) [20]. This antioxidant capacity allows it to restore the activity of these enzymes; according to the chromatographic analysis, the ethanolic extract of *S. humilis* and the presence of α-amyrin-enhancing endogenous antioxidant capacity, induced by the Nrf2 pathway, can restore the action speed of the antioxidant enzymes [21]. To this effect, ursolic acid can be added as a non-enzymatic antioxidant capable of fortifying cellular and organismal antioxidant defenses, thereby mitigating oxidative stress.

Another source of the generation of reactive oxygen species in the kidney is the enzyme NADPH oxidase (NOX), specifically NOX4. It has been shown that, during the development of diabetes mellitus, the increase in the synthesis of renal angiotensin II contributes to the generation of superoxide anion radical in the kidney through NOX4 in mesangial cells and podocytes. When oxidative stress is generated through angiotensin II, it inhibits the expression of Nrf2, which explains why the expression of antioxidant enzymes decreases in the group of diabetic rats. Nevertheless, it can also cause damage to the kidney through profibrotic processes, in which the transforming growth factor beta (TGF beta 1) is one of the main factors that contribute to the profibrotic process. The presence of TGF beta 1 has also observed tubulointerstitial changes via proximal tubular cell hyperplasia, hypertrophy, and atrophy, which is consistent with the results observed in the present study [22]. Flavonoids and other phenolic compounds are known to reduce oxidative stress, reduce necrosis, regenerate β-cells, and have a nephroprotective effect.

In the presence of the ethanolic extract of *S. humilis*, a nephroprotective effect was observed. It has been pointed out that ursolic acid, quercetin, and gallic acid can protect the kidney by reducing oxidative stress and the inflammatory response, in which the presence of tumor necrosis factor-alpha (TNFα), interleukin 1-beta (IL-1B), and monocyte chemoattractant protein 1 (MCP-1) has been observed [23,24,25].

The generation of oxidative stress will not only cause cellular damage locally but also systemically in the vascular wall. NADPH oxidase is expressed through Nox1 and Nox2, which can generate the superoxide anion that can act on the synthesis of nitric oxide, leading to its inactivation by oxidation in its cofactor, increased degradation of nitric oxide synthesis, or changes in the expression of nitric oxide synthase, potentially causing endothelial dysfunction, in which a decrease in the synthesis of nitric oxide was observed, along with an increase in the activity of vasoconstrictor substances mainly angiotensin II, which supports the results found in our study [26]. The search for new compounds with hypotensive and antihypertensive activity can improve the quality for patients, so, in the present study, we decided to test whether the ethanolic extract of *S. humilis* had a vasorelaxant effect.

Previous studies carried out in our laboratory using the normotensive anesthetized rat model showed that the ethanolic extract of *S. humilis* decreased systolic blood pressure in a dose-dependent manner, since, in whole preparations, it is difficult to propose a mechanism of action due to the complexity of the mechanisms that regulate systolic blood pressure. Subsequent experimental series were carried out via in vitro preparations, and most of the vasorelaxation studies were performed using the rat aortic rings model, which is a small elastic artery poorly involved in the regulation of arterial blood pressure compared to small resistance arteries [27]. In a study on vascular reactivity in the aortic rings of normotensive rats with and without vascular endothelium, pre-contracted with α1-adrenoceptor agonist phenylephrine, depolarization with high K^+^ solution was found to induce increases in smooth muscle tone by stimulating Ca^2+^ influx through voltage- and receptor-operated Ca^2+^ channels [28]. In both experimental series, a dose-dependent vasorelaxing effect was observed.

To determine whether the vasorelaxant effect of ethanolic extract of *S. humilis* is dependent on nitric oxide, the main vasodilatory compound involved in the relaxation of smooth muscle cells, the rings were preincubated with the nitric oxide synthase inhibitor L-NAME. As shown in Figure 6, we observed that the response was similar in the presence and absence of endothelium, suggesting that the vasorelaxant effect is independent of nitric oxide and the vascular endothelium. According to the analysis of the secondary metabolites present in the ethanolic extract, the presence of oleanolic acid, acid gallic, quercetin, and α-amyrin was found. Previous studies have shown that it has a vasorelaxant effect independent of the endothelium [29].

During the contraction process in the smooth muscle cell, the increase in intracellular Ca2+ plays an important role. The aortic rings were preincubated with a calcium channel blocker verapamil, and in the presence and absence of the ethanolic extract of *S. humillis*, the concentration–response curve was observed again. In the presence of verapamil, relaxation was reduced in a dose-dependent manner, suggesting that the vasorelaxant effect of the ethanolic extract of *S. humillis* was induced through the blockade of calcium channels (Figure 7).

## 4. Materials and Methods

### 4.1. Preparation and Identification of the Ethanol Extracts of Swtenia Humilis

The fresh seeds of *Swietenia humilis* were collected in March 2019 at Sonora, Mexico (N 19°15″, W 99°06′00″), and authenticated with a voucher specimen of *S. humilis* (3456) from Facultad de Estudios Superiores Iztacala, Universidad Nacional Autónoma of México (UNAM).

In traditional medicine, to efficiently extract secondary metabolites, especially flavonoids, they must be macerated in hydroethanolic solutions at ethanol concentrations of 30 to 70%; due to their low toxicity for human consumption, this is considered a safe and efficient extraction procedure. Therefore, in the present study, we decided to use the maceration method, which consisted of placing fresh seeds of *S. humilis* (1 kg) extracted twice with ethanol (3 L) at room temperature for 14 days and concentrating them in a rotary evaporator (Buchi Rotavapor model Mp60, Flawil, Switzerland) under reduced pressure to obtain a final volume of 50 mL. The solvent was completely removed when the extract was dried. Ethanol extracts of *S. humilis* were stored at 4 °C.

### 4.2. Identification of Compounds by High-Performance Liquid Chromatography (HPLC)

The ethanol extracts were analyzed using a Hewlett Packard 1100 series high-performance liquid chromatograph (HPLC) with an autosampler (Agilent Technologies, Santa Clara, CA, USA, 1200 series) and a diode array detector Hewlett Packard 1100 series.

For the flavonoids, the analyses were performed on a Hypersil ODS (125 × 40 mm) Agilent Technologies column with a gradient of (A) H_2_O at pH 2.5 with trifluoroacetic acid (TFA) and (B) acetonitrile (ACN) for 0–0.1 min with 85% solution A and 15% solution B, 0.1–20 min with 65% solution A and 35% solution B, and 20–25 min with 65% solution A and 35% solution B, as well as under the following parameters: flow—1 mL/min; temperature—30 °C; λ_1_—254 nm; λ—316 nm and 365 nm; analysis time—25 min. The standards used were apigenin, catechin, hesperidin, isorhamnetin, kaempferol, morina, myricetin, naringenin, phloretin, phloridzin, quercetin, and rutin.

For phenolic acids, a Nucleosil column (125 × 4.0 mm) from Macherey-Nagel (Düren, Germany) was used with a gradient of (A) H_2_O at pH 2.5 with trifluoroacetic acid (TFA) and (B) acetonitrile (ACN) for 0–0.1 min with 85% solution A and 15% solution B, 0.1–20 min with 65% solution A and 35% solution B, and 20–23 min with 65% solution A and 35% solution B. The other experimental parameters included the following: flow—1 mL/min; temperature—30 °C; λ—254, 280, and 330 nm; and analysis time—25 min. Caffeic, chlorogenic, ferulic, gallic, p-coumaric, p-hydroxybenzoic, protocatechuic, resorcilic, rosmarinic, sinapic, syringic, and vanillic acids were used as the standards.

The terpenoids were analyzed with a Zorbax Eclipse XDB-C8 column (Agilent, Maynooth, Kildare, Ireland) (125 mm × 4 mm, 5 μm). All the constituents were separated via isocratic analysis, using a mobile phase with 20% water and 80% acetonitrile, the flow was 1 mL/min for 21 min; the detector was adjusted to 215 and 220 nm [30]. α-amyrin, carnosol, oleanoic acid, β-sitosterol, stigmasterol, and ursolic acid were used as the standards.

External standards (Sigma Co., Kawasaki, Japan) were used to identify and quantify each of the flavonoids, phenolic acids, or terpenoids mentioned above; the solutions of pure compounds were prepared at concentrations of 0.08, 0.16, 0.32, 0.64, and 1.28 mg/mL in HPLC-grade methanol. Using readings from each series of standard solutions for areas of peak absorption and flavonoid, phenolic acid, or terpenoid concentration, linear regression equations were obtained to calculate the content of the compounds in the samples.

### 4.3. Antioxidant Activity In Vitro

The determination of phenols by Singleton was carried out using the colorimetric oxidation-reduction reaction with the Folin–Ciocalteu reagent (Millipore Sigma, Burlington, MA, USA) as an oxidizing agent and with a 0.2 mg/mL gallic acid curve. The samples were incubated for 30 min at 37 °C, and the absorbance was determined at 760 nm. The DPPH radical scavenging assay was carried out in 96-well plates, to which 50 µL of different concentrations of the extracts (10 to 100 µg/mL) and 150 µL of methanolic DPPH solution (250 µg/mL) were added in triplicate, in µM. Quercetin was used as the reference standard. The plates were incubated in the dark at 37 °C for 30 min. The absorbance was determined at 515 nm in an ELISA reader (Multiskan FC, Thermo Scientific, Singapore) [30].

#### 4.3.1. Markers of Oxidative Stress

##### Total Antioxidant Capacity

Total antioxidant capacity (TAC) was evaluated according to the method described by Pérez-Torres (2023) [31]. A total of 1 mL of blood plasma was suspended in 1.5 mL buffer composed of 20 mM of Cl_3_FeH_12_O_6_, 300 mM of NaC_2_H_3_O_2_, and 10 mM of 2,4,6-Tris-2-pyridil-s-triazine dissolved in 40 mM of HCL at pH 3.6. The absorbance was measured at 593 nm.

##### Malondialdehyde and Malonate

Malondialdehyde and malonate were determined simultaneously in the treated samples by capillary zone electrophoresis. The sample was diluted 1:2 with cold 0.1 M sodium hydroxide and analyzed directly. P/ACE MDQ system (Beckman Coulter, Urbana, IL, USA) was used, in which the capillary was preconditioned by passing a 0.1 M sodium hydroxide solution at 20 psi for 10 min, followed by deionized water for 10 min and, finally, the running buffer (10 mM borates + 0.5 mM CTAB at pH 9.0) for 10 min. The samples were injected under hydrodynamic pressure at 0.5 psi/10 s. Separation was performed at −25 KV for 4 min at 267 nm. The capillary was washed between each run at 20 psi with 0.1 M NaOH for 2 min, deionized water for 2 min, and running buffer for 4 min. Malondialdehyde and malonate concentrations are expressed in pmoles/mL and determined separately using a standard curve [32].

##### 8-Hydroxy-2-Deoxyguanosine

8-Hydroxy-2-deoxyguanosine was determined in the treated samples by capillary zone electrophoresis. The sample was deproteinized with 20% trichloroacetic acid, in a ratio of 10:1. It was centrifuged at 16,000× *g* (Spentrafuge 24D, Labnet, Palo Alto, CA, USA) for 15 min, filtered with 0.22 μm nitrocellulose membrane filters (Millipore, Darmstadt, Germany), diluted 1:10 with 2 M sodium hydroxide, and analyzed directly with P/ACE MDQ system (Beckman Coulter, Urbana, IL, USA), which was preconditioned with a capillary passage of 2 M sodium hydroxide solution for 30 min, and then deionized water for 30 min and, finally, the running buffer (10 mM borates at pH 9.0) for 30 min. The sample was injected under hydrodynamic pressure at 0.5 psi/10 s, and separation was performed at 20 kV for 8 min at 200 nm. The capillary was washed between runs with 2 M sodium hydroxide for 2 min and deionized water for 2 min. The results are expressed in pmoles/mL. The concentration of 8-hydroxy-2-deoxyguanosine was determined using a standard curve [31].

##### Antioxidant Activity In Vivo

Renal tissue catalase (CAT) activity was assayed at 25 °C, a method which is based on the disappearance of H_2_O_2_ from a solution containing 30 mmol/L H_2_O_2_ in 10 mmol/L potassium phosphate buffer (pH 7) at 240 nm. The glutathione peroxidase (GPx) activity was assayed following a previously described method. The results were expressed as UI/mg protein. Superoxide dismutase (SOD) activity in renal cortical homogenates was measured by a competitive inhibition assay using a xanthine–xanthine oxidase system to reduce NBT. The results were expressed as UI/mg protein [33].

### 4.4. Experimental Groups

For the diabetes mellitus model, male Wistar rats with an average weight of 250 g were used. They were provided by the Bioterium of the Facultad de Estudios Superiores Iztacala and were kept with free access to standard rat chow (Rodent Laboratory Chow 5001, Ralston Purina, St. Louis, MO, USA) and sterilized tap water, with 12–12 h light–dark cycles. Before inducing diabetes mellitus, plasma glucose was taken, and the blood sample was taken from the caudal end using a reflectance meter (One Touch; LifeScan, Milpitas, CA, USA). Subsequently, diabetes mellitus was induced by the intraperitoneal administration of streptozotocin (65 mg/kg of body weight in 10 mM sodium citrate buffer, pH 4.5); 15 min later, they were administered nicotinamide (110 mg/kg), and normoglycemic control (C) rats were administered only vehicle solution (10 mM sodium citrate buffer, pH 4.5). One week later, plasma glucose was taken again, and only animals with blood glucose levels >300 mg/dL were included in this study. The following groups were then formed:

Group 1: Normoglycemic control rats;

Group 2: Diabetic rats without treatment (DM);

Group 3: Diabetic rats treated with glibenclamide, 5 mg/kg, (DM + Gli);

Group 4 and 5: Diabetic rats treated with ethanolic extract of *S. humilis* 100 and 200 mg/kg (DM + EtOH 100 and 200 mg/kg).

Treatments were administered daily for 3 weeks before the animals were sacrificed. They were placed in metabolic cages to measure food and water consumption and urinary volume. The animals were then sacrificed and anesthetized with sodium pentobarbital (45 mg/kg, IP). Both kidneys were removed; the left kidney was used to determine the activity of antioxidant enzymes, while the right kidney was fixed with formaldehyde to make histological sections. Then, it was dehydrated using a graded series with ethanol, included in paraffin, sectioned in 5 mm thickness, mounted on slides, and stained with hematoxylin–eosin (HE).

Then, 1 mL of blood was used to determine angiotensin II and angiotensin 1-7 through capillary zone electrophoresis. The sample was deproteinized in a 1:1 ratio with cold 20% trichloroacetic acid. It was centrifuged at 16,000× *g* for 15 min at 10 °C (Sorvall RMC14, Dupont Inc, Wilmington, DE, USA) and filtered through 0.22 μm nitrocellulose membrane filters (Millipore, Darmstadt, Germany), diluted 1:2 with cold 0.1 M sodium hydroxide, passed through a cold Sep-Pak Classic C-18 cartridge (Waters Corporation, Milford, MA, USA), and analyzed directly with P/ACE MDQ system (Beckman Coulter, Urbana, IL, USA), to which the capillary was preconditioned by passing a 1.0 M sodium hydroxide solution for 30 min, and then deionized water for 30 min and, finally, the running buffer (100 mM boric acid + 3 mM tartaric acid + 10 fM gold III chloride at pH 9.8) for 30 min. The sample was injected under hydrodynamic pressure at 0.5 psi/10s. Separation was performed at 30 KV for 10 min at 200 nm at 20 °C. The capillary was washed between runs with 1.0 M sodium hydroxide for 2 min, deionized water for 2 min, and running buffer for 4 min. The results were expressed in pmoles/mL. Angiotensin II and angiotensin 1-7 concentrations were determined using a standard curve.

### 4.5. Preparation of Rat Aorta Rings and Vasorelaxant Effect

Normotensive male Wistar rats with an average weight of 250 g were anesthetized and sacrificed with sodium pentobarbital (50 mg/kg, i.p.). The thoracic aorta was removed, the surrounding tissue was removed, 4–5 mm long rings were cut, some rings were deprived of vascular endothelium by mechanical procedure, and each segment was placed on stainless steel hooks under an optimal tension of 3 g in an isolated organ chamber containing Krebs–Henseleit solution at (37 °C) and oxygenated (O_2_/CO_2_.95:5). The tension changes were recorded using Grass-FT03 force transducers (Astromed, West Warwick, RI, USA) connected to an MP100 analyzer (Biopac Instruments, Santa Barbara, CA, USA). After 60 min of stabilization, the rings were stimulated with noradrenaline (NA) (0.1 μM) for 10 min, and three solution changes were made each for 10 min to remove the stimulating agent. The absence of vascular endothelium was confirmed by the lack of relaxing response to carbachol (1 μM) after the last contraction; then, the aortic rings were precontracted with phenylephrine (1 × 10^−6^ M), and a concentration–response curve was made for the ethanolic extract of *S. humilis* at the following doses: 0.001, 0.0031, 0.01, 0.031, 0.1, 0.31, 1, and 3.1 mg/mL. In another experimental series, the aortic rings were precontracted with a nitric oxide synthase inhibitor (L-NAME), 10 μM. Once the record was stabilized, a concentration–response curve was made to the ethanolic extract, as previously described. Finally, to demonstrate that the ethanolic extract of *S. humilis* produces relaxation through the blockage of calcium channels, the aortic rings were subjected to a series of tests that were carried out in the experimental series. The rat aortic rings were incubated in a solution containing KCl 80 mM, the concentration–response curve was observed again in the presence of the ethanolic extract of *S. humilis*, and verapamil was used as a positive control. Muscle tone was calculated from the tracings using Acknowledge software (ver. 5.0.8, Biopac^®^ MP100 acquisition system, Goleta, CA, USA) [34].

### 4.6. Statistical Analysis

The data are presented as mean ± standard error of the mean. To determine the existence of significant differences between the treatments, the program used GraphPad Prism 5.0 software (GraphPad Software, Boston, MA, USA), treatments, and their inter-relationships by one-factor analysis of variance (ANOVA). When the inter-relationship and/or the main effects were considered statistically significant, the means were compared by employing Tukey’s test.

## 5. Conclusions

The ethanolic extract of seeds of *Swietenia humilis* showed antioxidant, hypoglycemic, and endothelium-independent vasorelaxant effects, probably by blocking calcium transport, ursolic acid, and α-amyrin, which are responsible for the effects observed in the present study.

## Figures and Tables

**Figure 1 ijms-26-02063-f001:**
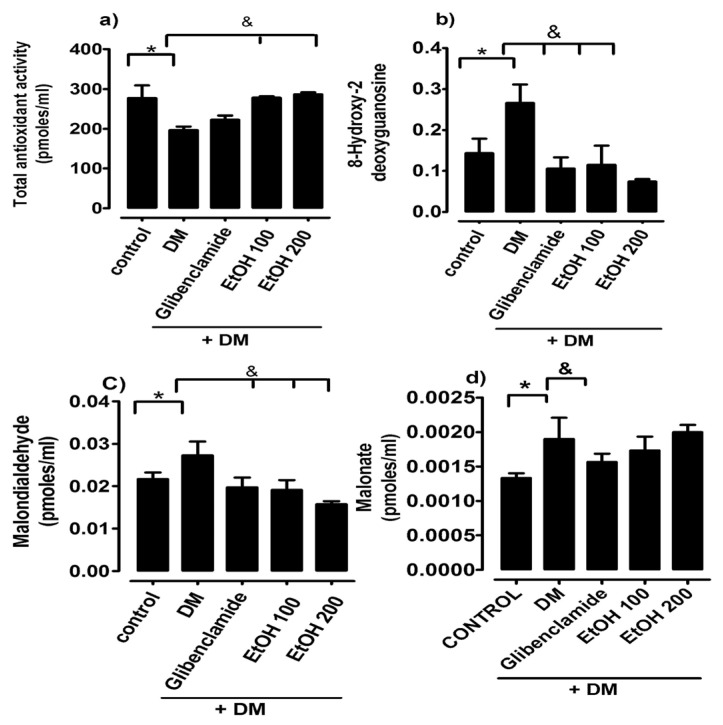
Comparison of total antioxidant activity (**a**), 8-hydroxy-2 deoxyguanosine (**b**), malondialdehyde (**c**), and malonate (**d**) in groups of rats: control (C), untreated diabetes mellitus (DM), diabetes mellitus treated with glibenclamide, and diabetes mellitus treated with ethanolic extract of *S. humilis*, after 4 weeks of treatment. The data are expressed as the average ± SEM. * C vs. treatments, *p* < 0.05. & untreated diabetes mellitus vs. treatments, *p* < 0.05.

**Figure 2 ijms-26-02063-f002:**
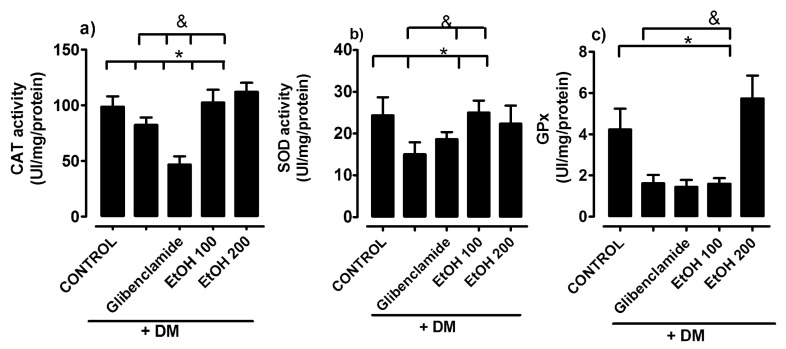
Antioxidant activity in the renal cortex catalase (**a**), superoxide dismutase (**b**), and glutathione peroxidase (**c**) in groups of rats: control (C), untreated diabetes mellitus (DM), diabetes mellitus treated with glibenclamide, and diabetes mellitus treated with ethanolic extract of *S. humilis*, after 4 weeks of treatment. The data are expressed as the average ± SEM. * C vs. treatments, *p* < 0.05. & untreated diabetes mellitus vs. treatments.

**Figure 3 ijms-26-02063-f003:**
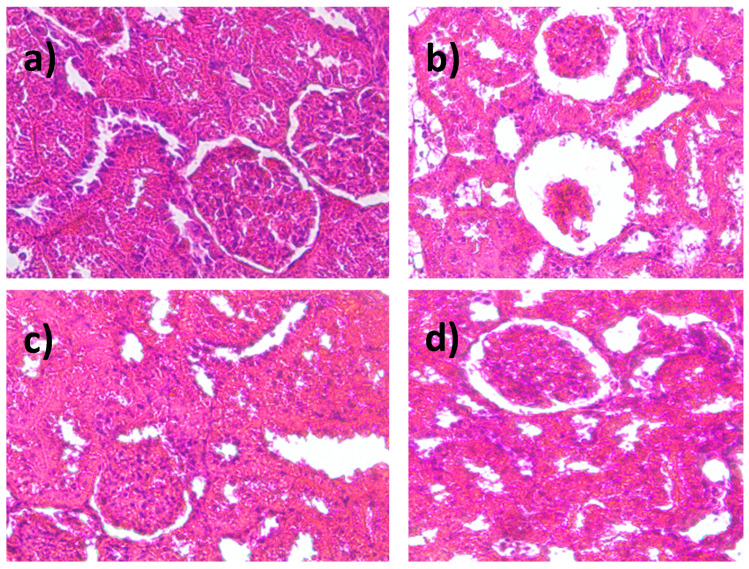
Photomicrography of histological sections of the kidney: (**a**) control; (**b**) diabetes mellitus (DM); (**c**) DM+glibenclamide; (**d**) DM + ethanolic extract of seeds of *S. humillis* (200 mg/kg).

**Figure 4 ijms-26-02063-f004:**
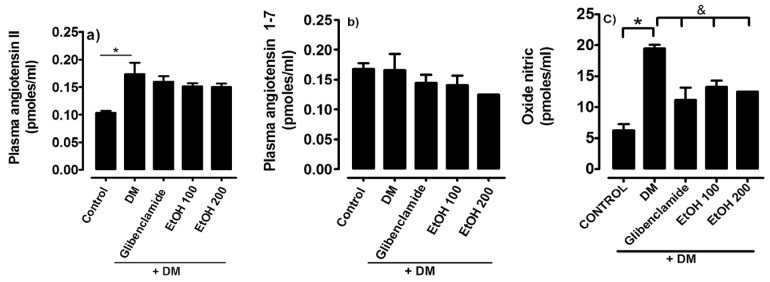
Plasma angiotensin II (**a**), plasma angiotensin 1-7 (**b**), and nitric oxide (**c**) in groups of rats: normoglycemic control (C), untreated diabetes mellitus (DM), diabetes mellitus treated with glibenclamide, and diabetes mellitus treated with ethanolic extract of *S. humilis.* The data are expressed as the average ± SEM. * C vs. treatments, *p* < 0.05. & untreated diabetes mellitus vs. treatments, *p* < 0.05.

**Figure 5 ijms-26-02063-f005:**
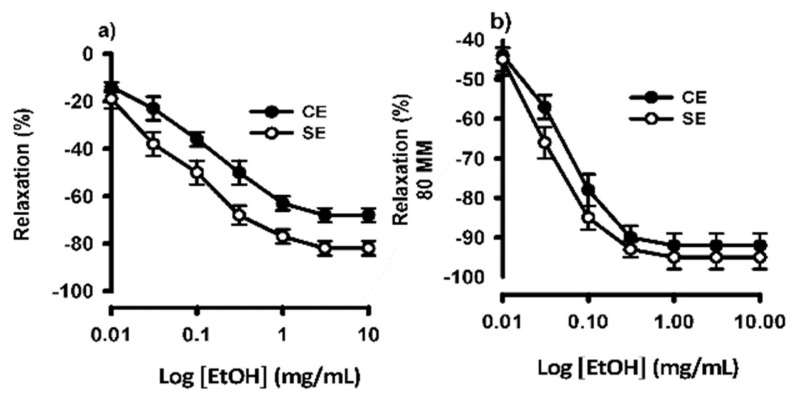
(**a**) Concentration–response curves of the ethanolic extract of *S. humilis* (EtOH) on phenylephrine-pre-constricted artery rings with or without endothelium, and (**b**) concentration–response curves of the relaxant effect of ethanolic extract of *S. humilis* on the contraction induced by KCl (80 mM) in artery rings with or without endothelium. Each point represents the mean ± S.E.M. of 3 animals.

**Figure 6 ijms-26-02063-f006:**
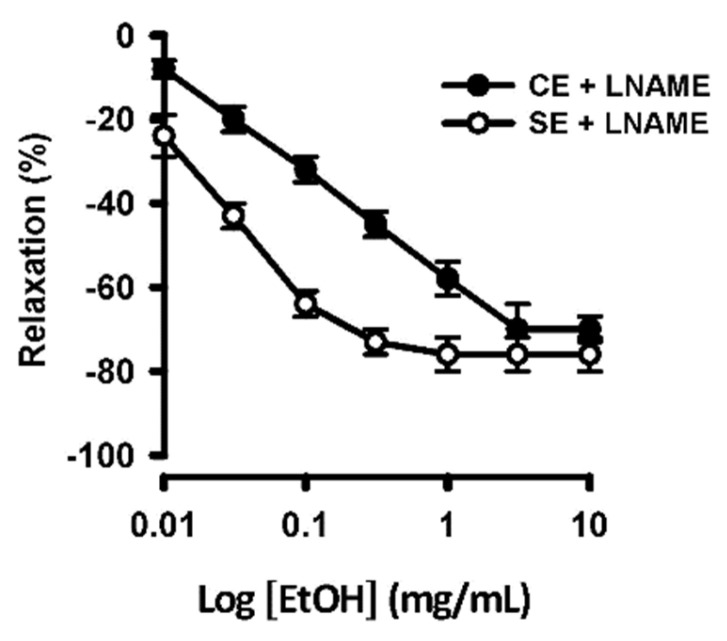
Concentration–response curves to ethanolic extract of *S. humilis* on unspecific nitric oxide synthase inhibitor (L-NAME)-pre-constricted artery rings with or without endothelium. Each point represents the mean ± S.E.M. of 3 animals.

**Figure 7 ijms-26-02063-f007:**
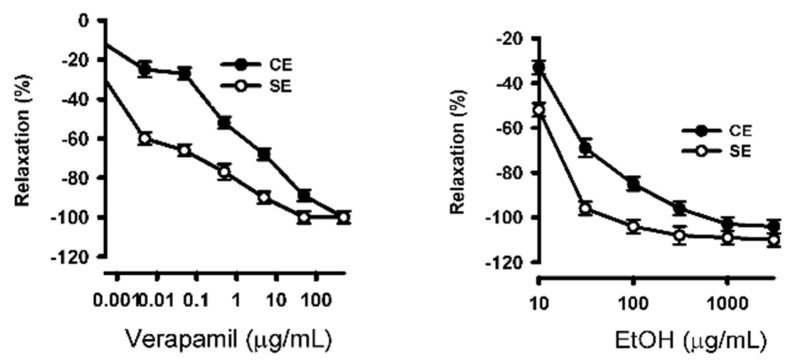
Ethanolic extract of *S. humilis* promoted vasodilation by L-type calcium channels. Concentration-constriction curves show the effect of ethanolic extract of *S. humilis* pre-incubation on high K^+^-constricted aortic rings.

**Table 1 ijms-26-02063-t001:** Phytochemicals detected in ethanolic extract of *S. humilis*.

	Retention Time (min)	Area (mAU*s)	Variables	%
1	4.705	60.56194	rutin	0.0623
2	7.409	299.36691	myricetin	1.2591
3	11.533	153.40312	quercetin	0.2781
4	13.021	305.80054	floretin	1.6702
5	21.687	565.50159	galangin	1.2500
6	2.672	35.75129	gallic acid	0.0205
7	4.985	36.1048	syringic acid	0.0407
8	9.184	9.21700	ferulic acid	0.1485
9	10.226	38.14323	p-coumaric acid	0.016
10	2.370	2048.18	carnosol	0.372
11	4.410	423.059	oleanolic acid	0.938
12	2.651	1230.902	ursolic acid	12.454
13	6.740	422.363	α-amyrin	68.305

**Table 2 ijms-26-02063-t002:** Comparison of glycemia, body weight, water and food ingestion, and urinary volume in control (C), untreated diabetes mellitus (DM), diabetes mellitus treated with glibenclamide (DM + Gli), and diabetes mellitus treated with ethanolic extract of *S. humilis* (EtOH 100 y 200 mg/kg). The data are expressed as the average ± SEM. * C vs. treatments, *p* < 0.05. & untreated diabetes mellitus vs. treatments.

Parameter	Control	DM	DM + Gli (5 mg/kg)	DM + EtOH (100 mg/kg)	DM + EtOH (200 mg/kg)
Glycemia (mg/dL)	100 ± 8	480 ± 19 *	150 ± 2 *	425 ± 8 *&	182 ± 10 *&
Body weight (g)	366 ± 24	255 ± 8	300 ± 15 *	300 ± 15 *	338 ± 13 *
Food intake (g)	16 ± 2	40 ± 8 *	30 ± 3 *	36 ± 7	20 ± 5 *
Water ingestion (mL/24 h)	24 ± 3	75 ± 12 *	78 ± 4	62 ± 13	72 ± 8
Urinary volume (mL/24 h)	13 ± 4	50 ± 10 *	30 ± 2 *	56 ± 8	53 ± 6

## Data Availability

Data is contained within the article.

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
