# Peer review of "Antioxidant, Antidiabetic, and Vasorelaxant Effects of Ethanolic Extract from the Seeds of Swietenia humilis"

_ijms, 2025, doi:10.3390/ijms26052063_

Round 1
Reviewer 1 Report
Comments and Suggestions for Authors
Dear authors,
Your work presented in this manuscript focuses on the investigation of the antioxidant, antidiabetic and vasorelaxant effects of some ethanolic extracts of Swietenia humilis.
The subject is of interest, considering the number of patients suffering from diabetes in the world and of different types of diseases related to the oxidative stress.
Unfortunately, your paper is extremely difficult to read and follow. The writing is not correct, there are many sentences without sense, the data are not well organized and presented.
After carefully reading the manuscript, I have the following comments and suggestions:
- delete the point after the title
- rephrase lines 26 and 27
- line 31: add "the" antioxidant...
- line 44: add "the" pro-oxidant...
- line 47: avoid the repetition of "treatment"
- line 64: add "the" antioxidant...
- line 65: add "of the" ethanolic...
- line 69: use the correct symbol for alpha
- line 74: eliminate the point after the table's title
- in Table 1, use the correct symbol for alpha
- line 76: "the" phenols,
- line 77: "the" average
- rephrase lines 76-78, the sentences make no sense
- rephrase lines 84-85, the sentences make no sense
- eliminate "they" in line 86
- line 95: add "it" can be seen..
- line 95: what do you mean by "the total antioxidant activity"?
- redo the Figure 2. The column with diabetes is not marked. What is the significance of the symbol "&" in the Figure? The notations in Figure do not correspond with those in legend
- rephrase lines 112-116, the sentences make no sense
- starting from line 195, there is a different font size
- the Conclusion part needs to be more detailed
- the Conclusion part appears 2 times in the manuscript
- how is the antioxidant activity evaluated? it is not clear what the aborbances read serve for
The references are not updated.
Comments on the Quality of English LanguageUnfortunately, your paper is extremely difficult to read and follow. The writing is not correct, there are many sentences without sense, the data are not well organized and presented.
Please see my comments.
Author Response
- Unfortunately, your paper is extremely difficult to read and follow. The writing is not correct, there are many sentences without sense, the data are not well organized and presented.
For the review of the observations noted, they were marked in purple for easy identification in the text.
The wording of the article was corrected for better understanding according to the observations made, the introduction was modified, the results section was also reorganized for a better understanding of the data. The discussion was reorganized and modified according to the findings made. The methodology was also modified
- redo the Figure 2. The column with diabetes is not marked. What is the significance of the symbol "&" in the Figure? The notations in Figure do not correspond with those in legend
In all figures, the legend of the group with diabetes was added and the figure caption was also modified.
3. The Conclusion part needs to be more detailedThe conclusion was modified
4.The references are not updated.References were updated
5. how is the antioxidant activity evaluated? it is not clear what the aborbances read serve for In vivo the antioxidant activity was evaluated, through capillary zone electrophoresis, one milliliter of blood plasma used, was suspended in 1.5 mL buffer composed of 20 mM of Cl3FeH12O6, 300 mM of NaC2H3O2, and 10 mM of 2,4,6-Tris-2-pyridil-s-triazine dissolved in 40 mM of HCL at pH 3.6. The absorbance was measured at 593 nm, Malondialdehyde and 8-Hydroxy-2-deoxyguanosine were determined as markers of oxidative stress.In tissue samples, the renal cortex was used to determine the antioxidant activity of the enzymes catalase, superoxide dismutase and glutathione peroxidase.In vitro the antioxidant activity In the ethanolic extract the concentration ofphenols by Singleton was carried out using the colorimetric oxidation-reduction reaction with Folin-Ciocalteu and The antioxidant capacity was determined by measuring the percentage of decolorization of the radical (2,2-diphenyl-1-picryl-hydrazyl). DPPH absorbs at a maximum of 517 nm when reacted with an antioxidant compound
6. what do you mean by "the total antioxidant activity"?
The total antioxidant activity is due to the participation of the following molecules endogenous antioxidants such as albumin, bilirubin, glutathione (GSH), and uric acid; antioxidant enzymes, such as superoxide dismutase (SOD), catalase (CAT), glutathione peroxidase (GPx), heme oxygenase-1 (HO-1), and NAD(P)H: quinone oxidoreductase (NQO1), and in order to quantify them we used blood plasma and they were determined by capillary zone electrophoresis. A paragraph was added to the methodology where the procedure followed to quantify the total antioxidant activity is described.

Reviewer 2 Report
Comments and Suggestions for Authors
The article "Antioxidant, Antidiabetic, and vasorelaxant effects of ethanolic extract from the seeds of Swietenia humilis" is well-written and well-presented. It discusses an interesting topic and investigates the efficiency of the Swietenia humilis ethanolic extract on diabetes and some other biological parameters.
However the manuscript needs improvement, here are some suggestions.
1- The authors didn't pay sufficient attention to phytochemistry throughout the article. This flaw could be noticed in the introduction which is not well-developed and did not include the reported plant's chemical compositions. Additionally, they didn’t discuss the chemical classification of the detected compounds in the dedicated section.
2- The authors included a long introductory part in the discussion section. It would be better if they merged lines 182 to 228 with the introduction,
3- There is no legend for Figure 1, moreover, compounds 5-13 have no clear peak on the chromatogram. The authors should clarify how they detected compounds in these regions (If they were identified by mass, this should be depicted in the figure!).
4- The author didn't clarify the applied method for the chemical profiling (standards, a database, or a bibliography?).
5- There was no justification in the study for the selection of ethanol for extraction. Was it because of the nature of the extracted compounds (as there are phenolics) or due to the low toxicity, etc.?
6- The authors didn't correlate the polyphenolic structure of the detected compounds and the biological activity. They didn't mention any of their reported activity as antioxidant and antidiabetic in the literature (e.g. quercetin is reported as antidiabetic, most of the listed phenolics are well-known as antioxidants, etc.)!
7- The used font should be unified, (lines 195-265)
8- The conclusion should be enriched, it was also duplicated in sections 3.1 and 5.
This article can't be published in its current form, it needs deep modifications to be acceptable.
Author Response
1.The authors didn't pay sufficient attention to phytochemistry throughout the article. This flaw could be noticed in the introduction which is not well-developed and did not include the reported plant's chemical compositions. Additionally, they didn’t discuss the chemical classification of the detected compounds in the dedicated section.
The modifications made to the document were marked in purple for easy identification. The introduction and discussion were modified according to the observations made.
2.The authors included a long introductory part in the discussion section. It would be better if they merged lines 182 to 228 with the introduction
Suggested text added to the introduction
3.There is no legend for Figure 1, moreover, compounds 5-13 have no clear peak on the chromatogram. The authors should clarify how they detected compounds in these regions (If they were identified by mass, this should be depicted in the figure!).
The legend was added to figure 1, one of the most abundant compounds found in this study was alpha amirin, so the peak was higher and with respect to those of lesser abundance in which they indicate 5-13 these peaks were almost flattened.
4. The author didn't clarify the applied method for the chemical profiling (standards, a database, or a bibliography?).
For the identification of secondary metabolites, the equipment has a library of 14 standards for phenolic acids and 12 for flavonoids. The methodology section addresses which these standards are according to the metabolites to be determined.
5. There was no justification in the study for the selection of ethanol for extraction. Was it because of the nature of the extracted compounds (as there are phenolics) or due to the low toxicity, etc.
In traditional medicine it is mentioned that in order to efficiently extract secondary metabolites, especially flavonoids, they must be macerated in hydroethanolic solutions at ethanol concentrations of 30 to 70%, due to their low toxicity for human consumption, which is why it is considered a safe and efficient extraction procedure, so in the present study it was decided to use the maceration method which consisted of placing fresh seeds of S. humilis were extracted twice with ethanol
6.The authors didn't correlate the polyphenolic structure of the detected compounds and the biological activity. They didn't mention any of their reported activity as antioxidant and antidiabetic in the literature (e.g. quercetin is reported as antidiabetic, most of the listed phenolics are well-known as antioxidants, etc.)!
In the discussion section this corresponding analysis was carried out according to the results obtained.
7. The conclusion should be enriched, it was also duplicated in sections 3.1 and 5.
the conclusion was modified

Round 2
Reviewer 1 Report
Comments and Suggestions for Authors
Dear authors,
Thank you for considering my comments and suggestions and for making the changes required in the manuscript.
I suggest you detail the Conclusions part.
Author Response
The conclusion was modified
some texts of the document were modified for better understanding
the changes are marked in green for easy identification.

Reviewer 2 Report
Comments and Suggestions for Authors
Thank you for the modifications, which improved the clarity and presentation of the article. However, the conclusion is still poor, and Figure 1 doesn't introduce beneficial information. I suggest deleting it, as Table 1 is more informative. This would make the article more concise.
Comments on the Quality of English Language
No comments
Author Response
The conclusion was modified
The figure was removed based on the suggestions made
Some sections of the writing were modified for better understanding, they are marked in green for easy identification
